# OpenReview forum: "CoopGuard : Cooperative Agents Safeguarding LLMs Against Evolving Adversarial Attacks"
_ICLR.cc/2026/Conference — ICLR 2026 Conference Withdrawn Submission_

### Official Review · Reviewer_xxxF · 2025-10-22

**Soundness:** 1
**Presentation:** 3
**Contribution:** 2
**Rating:** 2
**Confidence:** 4

**Summary:**

This work proposes a defense against "evolving" adverarial attacks.
Here, "evolving" refers to a multi-round setting in which an attacker interacts with a defending model and may incrementally craft new attacks based on the model's responses (even non-affirmative ones).
The authors argue that such a threat model necessitates an "adaptive" defender that can take previous rounds of interaction into account and actively try to disrupt the attacker.

To implement this idea, the authors propose a multi-agent system (with each agent being depended on the entire query history) composed of (1) a "deferring agent" that computes a scalar attack detection score, (2) a "tempting" agent that is prompted to generate misleading responses, (3) a "forensic agent" that provides textual summaries of the interaction history and (4) a system agent that orchestrates these individual agents to compute a defense policy.

To evaluate their defense, the authors pre-compute a dataset of $N \times 52$ input sequences and $N$ target sequences, with each element being composed of an original query, rephrased query, and 50 different jailbreak-style modifications of the original query.
The effectiveness in defending against these attacks is evaluated by (1) comparing to the target sequence (2) using an LLM judge (3) tracking the number of tokens needed for attacks.
The method compares favorably to prior, non-adaptive defenses.

**Strengths:**

* The main conceptual contribution, which is that defending against multiple round of attacks in an interaction requires an adaptive/active defender is sound and could be of interest to the LLM robustness community
* The discussion of related work is extensive
* The authors acknowledge that simply evaluating attack success based on matching with a target sequence is inadequate, which they address by also using an LLM judge
* They also acknowledge that LLM judges themselves may be unreliable, and thus conduct experiemnts of their reliability in Appendix E, which may be of independent interest

**Weaknesses:**

### Main weaknesses
The experimental evaluation is methodologically deeply flawed, ignoring well-established (and repeatedly learned) lessons in robustness literature.
This work proposes an empirical adverarial defense. Soundly evaluating empirical defenses is crucially dependent on using **adaptive attacks** (not to be confused with the adaptive defense discussed in this work). That is, the defense needs to be evaluated against attacks that are specifically crafted to circumvent it
(e.g., by backpropagating through the entire defended model), to provide at least some assurance that the defense will not be trivially circumvent within the next few weeks / months [1, 2].
In contrast **the proposed evaluation is the antithesis of an adaptive attack**. It uses a set of pre-computed queries that are not taking into account the agentic nature of the defender.
I understand that this is flavor of evaluation is common within the LLM robustness community, but this does not make the evaluation any less flawed (for a recent demonstration of the importance of adaptive attacks in the LLM context, see also [3]).

Even ignoring the issue of non-adaptivity, the **evaluation itself seems mismatched/orthogonal to the "evolving" threat model** that is used to motivate the proposed defense. The entire reason for using "delaying" or "tempting" responses is that they may inhibit the "**evolving**" attacker. However, the entire dataset / set of adversarial interactions is **pre-computed** before any interaction happens. Thus, the attacker has no chance of being influenced by the defender.

### Other Weaknesses
* While the general idea of using a stateful defender for the "evolving" threat model is sound and well-founded, the concrete instantiation via "CoopGuard" is not justified and instead relies simply on  anthropomorphization of different agents
* It is not clear why the chosen set of agents is sufficient, and we shouldn't be introducing even more agents.
* Conversely, no ablation study is conducted. It is not clear why using fewer agents / only using the "system coordinator" wouldn't be sufficient.
* While the authors are to be commended for evaluating the LLM judge itself, the evaluation seems insufficient.
In particular, no adaptive attack is conducted against the entire CoopGuard --> Judge pipeline.
* It is not clear how CoopGuard interacts with non-harmful prompts. Specifically, it is not clear if it passes non-harmful prompts to another LLM or generates responses itself.
* The discussion of limitations is relegated to Appendix L. This is a central part of the paper and I would strongly encourage the authors to move this to the mai ntext.
* The claim "this incremental attack pattern is more common" is not substantiated (e.g., using references to prior work)
* The dataset (just 5200 prompts) seems incredibly small compared to the parameter count of the models, and thus ill-suited for gauging the population-level adversarial risk.

### Minor comments
* I would suggest replacing the term "independent attack". At first glance it seems to contradict the idea of "attack attempts that become progressively refined" (l.78). I understand that "independent" is meant to refer to multiple rounds of interaction instead of "a unified jailbreak sequence" (l.077). But taken by itself, it  falsely suggests that the interaction history did not matter (cf. independence in Markov chains).
* The submission claims that prior work "struggle[s] to generalize to novel adversarial tactics not seen during training" (l.76), but does not provide evidence that the proposed method offers better generalization.

**Questions:**

* How does CoopGuard interact with non-harmful prompts? Does CoopGuard itself generate the response, or forward to another model?

## Conclusion
Overall, this work makes a central conceptual contribution, namely defenses against "evolving" attacks requiring a stateful defender, which may of interest to the broader trustworthy ML community.
However, the proposed instantation of this idea is not motivated and does not provide any generalizeable insights for the reader.
Furthermore, the experimental evaluation completely ignores well-established practices for evaluating empirical defenses,
and the precomputed benchmark is ill-suited for evaluating robustness to the "evolving" attacks that motivated the work in the first place.
I therefore recommend rejection of this work

### References
[1] Athalye et al.Obfuscated Gradients Give a False Sense of Security: Circumventing Defenses to Adversarial Examples. ICML 2018
[2] Tramer et al. On Adaptive Attacks to Adversarial Example Defenses. NeurIPS 2020
[3] Nasr et al. The Attacker Moves Second: Stronger Adaptive Attacks Bypass Defenses Against Llm Jailbreaks and Prompt Injections. https://arxiv.org/abs/2510.09023

---

### Official Review · Reviewer_8pS4 · 2025-10-30

**Soundness:** 2
**Presentation:** 2
**Contribution:** 2
**Rating:** 4
**Confidence:** 3

**Summary:**

This paper introduces CoopGuard, a multi-agent defense framework designed to protect large language models (LLMs) from evolving, multi-round adversarial attacks. Unlike existing reactive or static defenses (e.g., RLHF-based or prompt-filtering approaches), CoopGuard deploys four cooperating agents — Deferring, Tempting, Forensic, and System — to delay, mislead, and analyze adversarial behavior dynamically. The authors also introduce a new benchmark, EMRA, comprising 5,200 adversarial samples across eight attack types that simulate independent but escalating attack rounds. Experiments on GPT-3.5-turbo, GPT-4, and Gemini-1.5-pro show that CoopGuard reduces attack success rate (ASR) by up to 78.9%, improves deceptive rate (DR) by 186%, and increases attack cost (token usage) by 167.9% over prior defenses such as PAT, RPO, GoalPriority, and Self-Reminder.

**Strengths:**

The paper introduces a novel cooperative multi-agent framework for LLM safety, advancing beyond conventional single-model defenses. Its explicit modeling of evolving multi-round adversarial behavior, coupled with deception-oriented counterstrategies, offers a creative and rigorous conceptualization of adaptive defense mechanisms. The integration of distinct yet cooperative roles—Deferring, Tempting, Forensic, and System—constitutes an original contribution that reframes how safety interactions can be dynamically coordinated. The technical formulation is solid: Algorithm 1 formalizes the cooperative policy with feedback-driven parameter updates, and Equations (1)–(4) precisely define agent behaviors. Moreover, the adoption of semantic evaluation via GPT-Judge—as opposed to keyword-based refusal detection—enhances empirical validity. The accompanying EMRA dataset represents a meaningful benchmark contribution, capturing incremental adversarial escalation in a realistic manner. Overall, the paper is clearly presented, supported by illustrative examples, structured diagrams, and comprehensive comparisons. Ethical implications and reproducibility aspects are explicitly addressed, underscoring the paper’s scholarly maturity and practical significance for improving LLM robustness and moderation workflows.

**Weaknesses:**

Despite its strengths, the empirical evaluation is somewhat limited in scope. Experiments rely mainly on GPT-based closed models, leaving questions about generalization to open-source systems such as LLaMA-3 or Mistral. Without such validation, the framework’s transferability and cost-effectiveness remain uncertain. The absence of ablation studies also weakens the causal interpretation of results—specifically, the contribution of each cooperative agent (e.g., Tempting or Forensic) is not quantitatively isolated. Additionally, the reliability of the GPT-Judge metric is underexamined; since it is a learned evaluator, alignment with human judgment or alternative safety scorers should be empirically verified. From a theoretical standpoint, the feedback loop described in Algorithm 1 (lines 11–12) lacks formal convergence or stability guarantees, relying instead on heuristic reasoning. Finally, scalability concerns persist: while qualitative claims of efficiency are made, no runtime or inference-cost comparisons against single-agent baselines are reported, leaving the real-world feasibility of multi-agent deployment somewhat ambiguous.

**Questions:**

1. It would be valuable to understand the individual contributions of each agent—Deferring, Tempting, and Forensic—to the overall reduction in ASR and DR. Could the authors consider including an ablation or breakdown experiment to quantify these effects?

2. Have the authors evaluated CoopGuard on open-source language models such as LLaMA-3-8B or Mistral-7B? It would be helpful to know how performance scales when applied to smaller or non-proprietary models.

3. To assess the robustness of the evaluation framework, did the authors conduct any human studies to verify that GPT-Judge’s harm and deception scores are consistent with human annotations or subjective judgments?

4. Could the authors provide details on the latency and token cost per defense round, particularly in comparison to baselines such as GoalPriority? This would help clarify whether CoopGuard is suitable for deployment in real-time or resource-constrained systems.

5. Finally, are there any insights into whether sophisticated adversaries could exploit predictable behavioral patterns from the Deferring or Tempting agents? Evidence of potential adversarial adaptation during extended interaction sessions would be of particular interest.

---

### Official Review · Reviewer_cdvp · 2025-11-01

**Soundness:** 1
**Presentation:** 2
**Contribution:** 2
**Rating:** 2
**Confidence:** 4

**Summary:**

The paper presents CoopGuard, a multi-agent system for defending adversarial attacks to LLMs. This multi-agent systems consists of 4 agents, a deferring agent, a tempting agent, a forensic agent and a system agent. This paper also presents a dataset EMRA designed to simulate multi-round attacks and to evaluate defense performance. Empirical results show that it achieves a stronger defense in terms of attack success rate and deceptive rate.

**Strengths:**

1. Very rich details provided in the appendices and I really appreciate that.
2. This paper is well organized and easy to follow.

**Weaknesses:**

1. The most critical fault of this paper is that it only evaluates the defense performance but does not evaluate the impact on benign or normal inputs. For example, will this defense lead to false positives that disturb a normal query. It is like only looking at the recall and ignoring precision, and solely improving the recall does not translate to a better performance. For example, a simple strategy to deny all request would lead to 0% attack success rate but it does not mean it is a good defense. Without benchmarking on false alarms on clean inputs, it is impossible to demonstrate the proposed method is better than others.

2. Another critical weakness is that it is lack of data or discussion around the defense cost and how it compares to other baseline methods. Intuitively such multi-agent system would be quite expensive so it is important to discuss the "defense efficiency" when compared with other methods.

**Questions:**

1. Since the deferring agent can evaluate the likelihood of an input prompt being jailbreak attack or not. What if we only use this agent to do this classification and deny response when it thinks it is an attack? What additional value other agents provide?

2. It seems EMRA is a static dataset. How a static dataset be able to perform multi-round ADAPTIVE attack? (line 308)

---

### Official Review · Reviewer_P4Km · 2025-11-03

**Soundness:** 2
**Presentation:** 2
**Contribution:** 2
**Rating:** 2
**Confidence:** 3

**Summary:**

The submission introduces CoopGuard – an agentic system designed to reduce jailbreak success rate – and a corresponding dataset EMRA that facilitates evaluation of CoopGuard. CoopGuard is designed to adapt to attacks that are themselves adaptive, and experiments show that it consistently obtains the lowest attack success rates on EMRA. Notably, CoopGuard is designed (and shown) to reduce attack efficiency as well as success rate, which may contribute to deterring attacks in practice.

**Strengths:**

**Originality** The approach to reducing attack efficiency is interesting and (as far as I know) novel. The EMRA dataset of attacks is also a new contribution.

**Quality** Experiments are conducted on a range of models and several strong baseline defenses are compared to, adding rigor to the analysis.

**Clarity** The paper is mostly clear and well-written.

**Significance** The overall structure of the approach is notable and introduces good ideas. For instance, the forensic analysis of attacks, the adaptivity of the defense strategy, and the honeypot-like tempting agent are all potentially influential ideas.

**Weaknesses:**

My understanding is that the evaluation uses relatively weak, static attacks from the proposed EMRA dataset. Given that the submission claims the proposed method CoopGuard addresses adaptive, multi-turn attacks, I would recommend evaluating on such attacks. In particular, the submission references methods like Crescendo, and there are many more such attacks (see "The Attacker Moves Second" by Nasr et al., 2025, for examples). These stronger, adaptive attacks would provide a clearer test of the claims made for CoopGuard.

**Questions:**

1. Line 27: a reduction greater than 100% is not possible, so please consider rephrasing this to avoid confusion.
2. Please discuss and compare to AegisLLM in the related work.
3. Please use parenthetical citations (\citep) where appropriate.
4. Line 251: should decoy usage rise as detection score (not "attacker confidence") rises?
5. Line 259: what's the difference between $L_{log}$ and $X_{1:t}$?
6. Is it correct to say that EMRA is two existing datasets (JBB and JDGP), combined with a GPT4 rephrasing of the first dataset?
7. Line 353: is there evidence to support usage of score=2 as an indicator of successful misdirection? Relatedly, if I understand Table 8 correctly, it indicates that humans find that CoopGuard is deceptive 54% of the time. Would it make sense to show the correlation between the human deceptive rating and the deceptive rating based on GPT-Judge score=2?
8. Line 175: How are these parameters updated? My understanding is that the agents are using GPT-4’s parameters.
9. Table 1: Why are only a subset of jailbreaking questions shown? Are performance trends similar for the other types of jailbreaking questions?
10. Table 1: What is the difference between “ASR (MR)” and “JQ (MR) ASR”?
11. Line 434: Is there evidence that longer interactions are provoked, or is this just a potential benefit of higher DR?

---

### Note · Authors · 2025-11-27

I have read and agree with the venue's withdrawal policy on behalf of myself and my co-authors.